# Linear time complexity de novo long read genome assembly with GoldRush

**Johnathan Wong** [1,2] ✉, **Lauren Coombe** [1,2], **Vladimir Nikolić** [1], **Emily Zhang**[1], **Ka Ming Nip** [1], **Puneet Sidhu**[1], **René L. Warren** [1,3] **& Inanç Birol** [1,3] ✉

Current state-of-the-art de novo long read genome assemblers follow the Overlap-Layout-Consensus paradigm. While read-to-read overlap – its most costly step – was improved in modern long read genome assemblers, these tools still often require excessive RAM when assembling a typical human dataset. Our work departs from this paradigm, foregoing all-vs-all sequence alignments in favor of a dynamic data structure implemented in GoldRush, a de novo long read genome assembly algorithm with linear time complexity. We tested GoldRush on Oxford Nanopore Technologies long sequencing read datasets with different base error profiles sourced from three human cell lines, rice, and tomato. Here, we show that GoldRush achieves assembly scaffold NGA50 lengths of 18.3-22.2, 0.3 and 2.6 Mbp, for the genomes of human, rice, and tomato, respectively, and assembles each genome within a day, using at most 54.5 GB of random-access memory, demonstrating the scalability of our genome assembly paradigm and its implementation.

Short-read genome assembly methods typically struggle to resolve sequence repeats and often fail to generate assemblies that reach chromosome-scale[1]. Both prokaryotic and eukaryotic genomes can contain a large proportion of repeats[2], with the human genome estimated to be 66–69% repetitive[3]. Thus, it is imperative that these repetitive regions be sufficiently resolved for a successful de novo genome assembly. Innovations in bioinformatics have emerged to address this challenge, leveraging long-range evidence afforded by various data types, including linked reads[4,5], Hi-C[6], and long sequencing reads[7,8].

Long-read sequencing technology has become increasingly prevalent in recent years. Sequencing throughput, affordability, and the long-read lengths are some of the key reasons[9]. The long-read lengths, ranging from kilobases to megabases, enable better resolution of structural variants[10] and long repeats[11]. Long reads also enable correct and accurate identification of tandem repeat expansions[12].

Oxford Nanopore Technologies (ONT), Plc. (Oxford, UK) and Pacific Biosciences (PacBio), Inc. (Menlo Park, USA) are currently the two preeminent providers of commercial long-read sequencing technology. PacBio generally produces long reads with lower base errors (8–13% and <1% for Continuous Long Reads (CLR) and HiFi,

respectively), but with shorter read lengths (typically averaging 30–60 kbp and 10–25 kbp for CLR and HiFi, respectively)[13,14] compared to that of ONT (typically 10–100+ kbp)[15]. Yet, high error rates (1–13%) in ONT reads remain a challenging obstacle to de novo genome assembly[15–17]. Unlike the de novo genome assembly strategies designed for short reads, both long-read sequencing technologies− and especially ONT, require algorithms and data structures that can accommodate mismatches and indels in the sequencing data.

Most long-read genome assemblers follow the Overlap-Layout-Consensus paradigm (OLC), a quadratic run time algorithm in its naïve implementation. OLC consists of three steps. The first step, overlap, typically generates an overlap graph by computing the pairwise alignment of all reads. As datasets often contain tens of millions of reads, finding and storing the detected overlaps is the most computationally- and memory-intensive step in the OLC paradigm, and has been the target of recent innovative algorithms[18–20]. In the second step, layout, the generated read overlap graph is traversed to produce contigs, or contiguous sequences, that reconstruct the underlying genome. The last step, consensus, uses read alignments to infer the most likely nucleotide bases across contigs, and corrects the sequences accordingly[21,22].

[1]Canada's Michael Smith Genome Sciences Centre, BC Cancer, Vancouver, BC V5Z 4S6, Canada. [2]These authors contributed equally: Johnathan Wong, Lauren Coombe. [3]These authors jointly supervised this work: René L Warren, Inanç Birol. ✉e-mail: jowong@bcgsc.ca; ibirol@bcgsc.ca

In recent years, a number of OLC-based de novo long-read genome assemblers have been developed that leverage the long-range evidence provided by the technology. These tools include Flye[18], Redbean[19], and Shasta[20]. Each tool brings a different innovation to the table, with implementations of the OLC paradigm aiming to reduce the computational cost and address the high error rates of long reads. For instance, Flye clusters the long reads that are likely to originate from the same genomic locus in a preprocessing step to reduce the number of pairwise comparisons[18]. Redbean segments each read into 256 bp tiling subsequences, reducing the dynamic programming matrix to a size of 65536 ( = 256 × 256), thus speeding up the pairwise alignment process[19]. On the other hand, to address the high error rate of long reads, Shasta compresses all homopolymers in the reads using run-length encoding, thereby removing all homopolymer expansion errors, one of the more common error types in the ONT data, and improving the accuracy of alignments in the overlap step of OLC[20]. While these optimizations have reduced the time it takes to assemble the long sequencing reads and ultimately improve upon the quality of the generated genome assemblies, these tools still have a large memory footprint, requiring upwards of several hundred gigabytes of RAM for assembling a typical 50X human genome dataset.

Long sequencing reads have also enabled the haplotype phasing of haploid genome assemblies to better understand and characterize the genomic diversity of diploid and polyploid organisms[23]. Existing OLC long-read genome assemblers, such as Shasta[24], and haplotype-aware genome assemblers, such as phasebook[25] and Verkko[26], have been extended or developed to leverage long reads to produce diploid assemblies.

In this work, we present GoldRush, a memory-efficient long-read haploid de novo genome assembler that employs a novel long-read assembly algorithm, which runs in linear time in the number of reads. GoldRush is implemented as a modular pipeline with four main steps: GoldPath, GoldPolish, Tigmint-long[7,27], and GoldChain[7,28]. GoldPath first iterates through the long reads and generates a "golden path", selected sequences with a ~1X representation of the genome of interest. Because the output of GoldPath is a set of raw sequences (termed "goldtigs"), base errors in the golden path are resolved using Gold-Polish, a long-read adaptation of the ntEdit+Sealer polishing protocol[29]. Next, misassemblies (due to chimeric reads) are corrected using Tigmint-long[7,27]. Finally, the corrected golden path is scaffolded using GoldChain to produce the output genome assembly[7,28] (Fig. 1a and Supplementary Fig. 1). The golden path serves as the base assembly for the subsequent steps in GoldRush. Briefly, GoldPath iterates through the reads, querying each read against a dynamic and probabilistic multi-index Bloom filter[30,31] (miBf) data structure in turn, and inserts selected sequence or skips over the read depending on the results of the query to generate multiple "silver paths", which are <1X representations of the target genome (Fig. 1b). The silver paths are then used as input for GoldPath to generate the golden path. Iteration over the long sequencing reads, as opposed to an all-vs-all alignment of reads, allows GoldRush to achieve a linear time complexity in the number of reads. We show that GoldRush produces contiguous and correct genome assemblies with a low memory footprint, and does so without read-to-read alignments, marking an important paradigm shift in the genome assembly of long sequencing reads.

## Results

We assembled the genomes of three different human cell lines (NA24385, HG01243, and HG02055), *Oryza sativa* (rice), and *Solanum lycopersicum* (tomato) using GoldRush, Flye, Redbean, and Shasta, and compared the resulting genome assemblies using a variety of length contiguity metrics, genome assembly accuracy markers, including those reported by QUAST[32], BUSCO[33], the asmgene utility in minimap2[34], and Merqury[35], and their resource usage (see Methods).

### Assembly performance

For the genome assemblies of all three human cell lines, GoldRush achieved NG50 lengths between 25.3 and 32.6 Mbp, comparable to both Shasta (29.7–39.6 Mbp) and Flye (26.6–38.8 Mbp), and typically three times more contiguous than the Redbean genome assemblies (8.0–10.9 Mbp) (Supplementary Tables 1–3). Two of the three human GoldRush genome assemblies (NA24385 and HG01243) also had the fewest extensive misassemblies (940 and 1057) among the tools tested; ~2–3 times fewer than Shasta (1682 and 3240), and ~5–7 times fewer than Redbean (4918 and 7052). Despite the relatively low number of structural misassemblies, the NGA50 length for each human GoldRush assembly is around 20 Mbp, indicating that some misassemblies are found in the larger scaffolds and breaking the alignment blocks (Fig. 2a and Supplementary Tables 1–3). In addition to assembling highly contiguous human genomes, GoldRush is also robust in assembling plant genomes, reaching 0.3 and 2.6 Mbp NGA50 lengths for *O. sativa* and *S. lycopersicum*, respectively (Fig. 2c and Supplementary Tables 4, 5). The *O. sativa* Shasta genome assembly, on the other hand, had scaffold NG50 and NGA50 lengths of 124,700 bp and 104,593 bp, respectively, only ~4.2 and ~3.6 times longer than the raw reads used as input (N50 = 29,349 bp) for genome assembly, respectively (Supplementary Tables 4, 6).

GoldRush and Shasta assembled each of the three human genomes in less than a day, executing in 16.6–20.8 h and 4.1–5.0 h, respectively (Fig. 2b and Supplementary Tables 7–9). Both Flye and Redbean required at least 33.7 h to assemble each of the three human genomes, with Redbean assembling two of the datasets (HG01243 and HG02055) in ~68 h, for each (Fig. 2b and Supplementary Tables 7–9). GoldRush is also competitive in assembling the smaller plant genomes, requiring 1.6 and 7.4 h to assemble *O. sativa* and *S. lycopersicum*, respectively (Fig. 2d and Supplementary Tables 10, 11). GoldRush used, at most, 54.5 GB of RAM to assemble the three human genomes (Fig. 2b and Supplementary Tables 7–9). In comparison, using the same data, Flye and Redbean used between 329.3–502.4 GB (six- to eight-fold more than GoldRush), and Shasta utilized 884.8–1009.2 GB (up to 20-fold more than GoldRush). Similarly, GoldRush required the least amount of RAM to assemble the *O. sativa* and *S. lycopersicum* datasets, using at most 45.3 GB (Fig. 2d and Supplementary Tables 10, 11).

### GoldPolish base error correction

GoldPolish decreased the number of mismatches or indels per 100 kbp of the NA24385 golden path by ~6.5-fold (1463.7 to 228.5 and 1327.2 to 197.2, respectively) (Supplementary Table 12). This improvement in mismatches and indels translated into a recovery of 12,272 (89.1%) complete BUSCOs, fewer than the 12,920 (93.8%) and 12,988 (94.3%) complete BUSCOs in the Shasta and Flye NA24385 genome assemblies, respectively, but more than the 12,193 (88.5%) complete BUSCOs reconstructed in the Redbean NA24385 genome assembly (Supplementary Table 13). Of the 2461 duplicated genes observed in T2T-CHM13[36], a complete telomere-to-telomere reference-grade human genome assembly, 845 were found in the GoldRush assembly in multiple copies, fewer than the 1717 and 1725 found in the Shasta and Flye genome assemblies, respectively, but greater than the 680 found in the Redbean genome assembly (Supplementary Fig. 2 and Supplementary Table 14).

When polishing the genome assemblies of the more erroneous human long-read datasets, HG01243 (estimated 9% error rate) and HG02055 (estimated 11% error rate), GoldPolish reduced the number of mismatches and indels per 100 kbp by 60.8 and 67.4% (to 1372.3 and 980.1) for the former, and by 53.6 and 60.5% (to 1981.7 and 1354.8) for the latter, respectively (Supplementary Tables 6, 15, 16). In comparison, the Flye, Redbean, and Shasta assemblies reported 148.7 and 110.6, 324.7 and 354.5, and 195.6

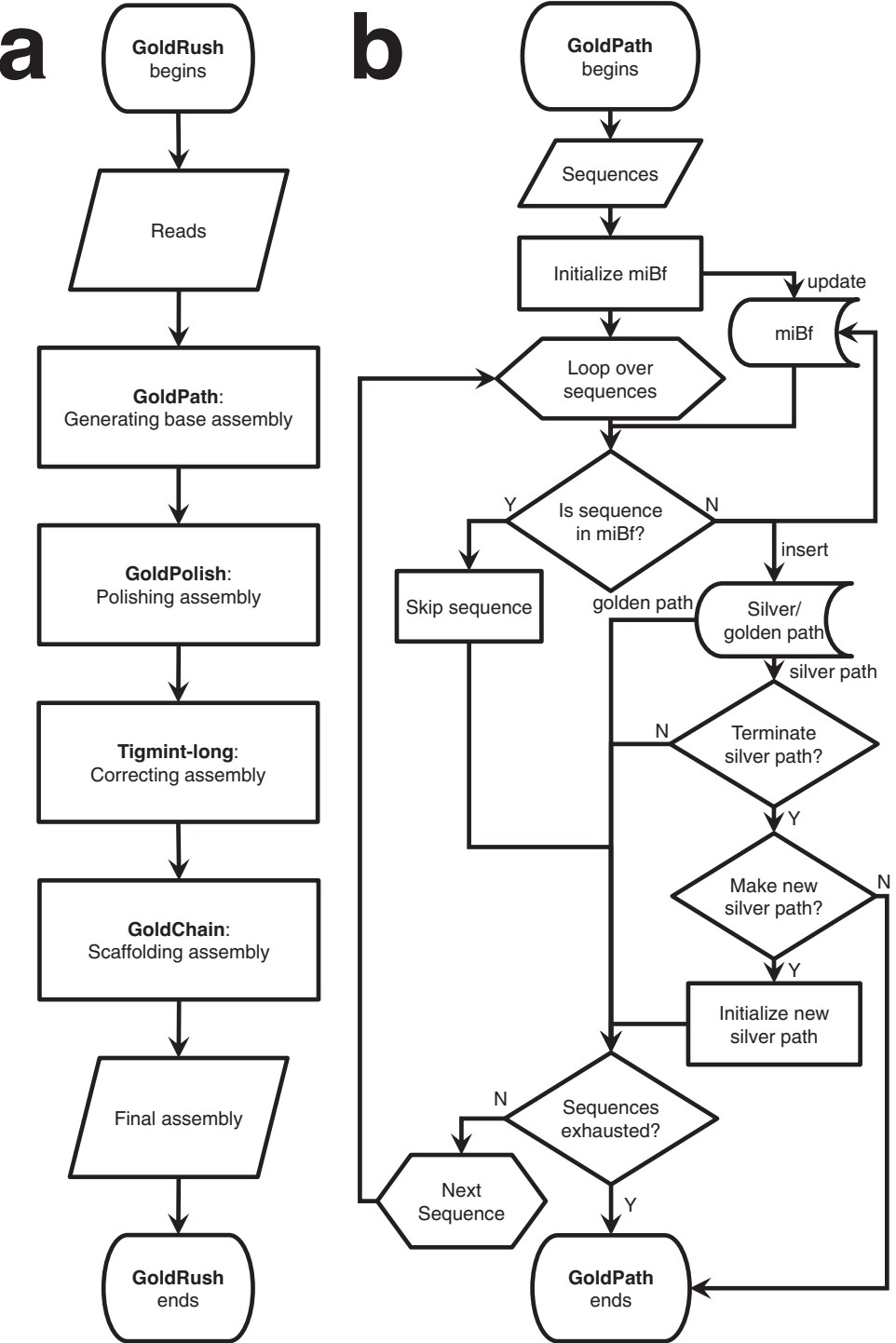

**Fig. 1 | Flowchart of GoldRush and GoldPath. a** Raw long reads are first processed by GoldPath to generate the golden path, a -1X representation of the genome. The golden path is then polished by GoldPolish and corrected for structural errors with Tigmint-long. Finally, GoldChain scaffolds the polished and corrected golden path to generate the final genome assembly. **b** GoldPath uses the input long sequencing reads or silver path sequences to initialize a miBf data structure. GoldPath then loops over the sequences, and queries each sequence against the miBf. If the sequence is found in the miBf, GoldPath skips it and resumes its iterations. Conversely, if the sequence is not found in the miBf, it is inserted into the miBf and added to the silver/golden path. When GoldPath is constructing a silver path, and if the silver path has not reached the threshold number of bases, GoldPath will continue recruiting bases from the input reads. If the threshold number of bases is reached, GoldPath will check if more silver paths need to be generated. If more silver paths are needed, GoldPath will create them using the same algorithm and parameters, otherwise, it will terminate. Five (by default) silver paths, each representing -0.9X (by default) coverage of the target genome, are combined to generate a low-coverage subsample input for GoldPath to build the golden path. When creating the golden path, GoldPath will continue iterating over the sequences from the silver paths until all sequences are exhausted.

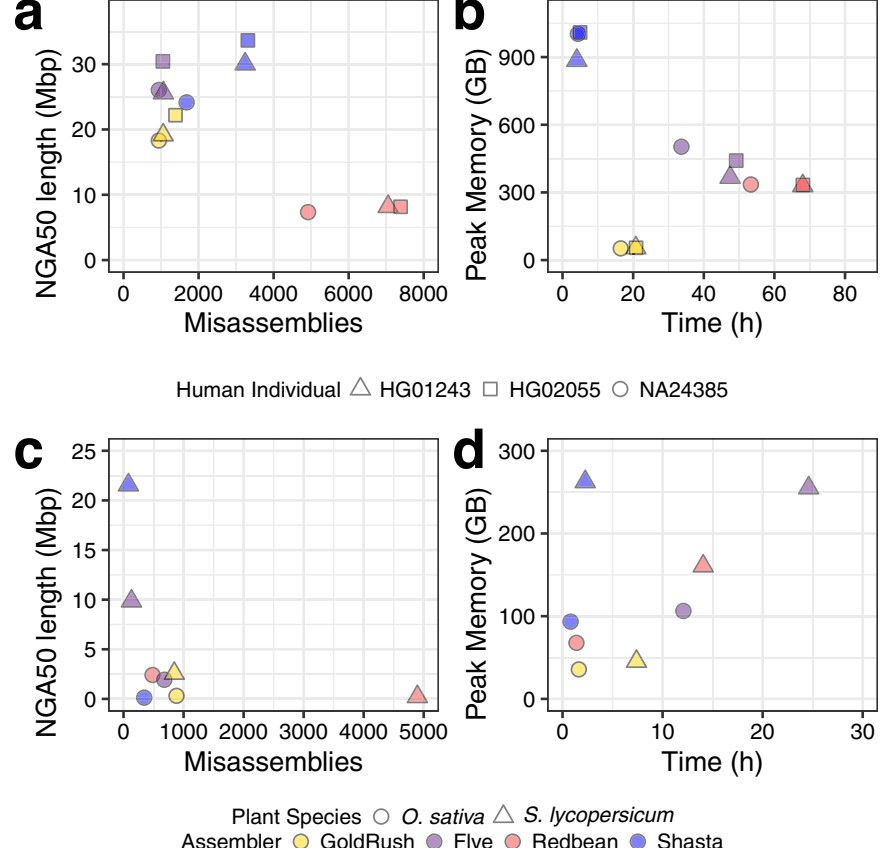

**Fig. 2 | Contiguity, correctness, and resource usage of GoldRush assemblies compared to Flye, Redbean, and Shasta genome assemblies for three human individuals (NA24385, HG01243, and HG02055), *O. sativa*, and *S. lycopersicum*.** Human individuals NA24385, HG01243, and HG02055 are represented as circles, triangles, and squares, respectively, in **a** and **b**. *O. sativa* and *S. lycopersicum* are represented as circles and triangles, respectively, in **c** and **d**. GoldRush, Flye, Redbean, and Shasta are colored yellow, purple, red, and blue, respectively. The genome assemblies were assessed using QUAST for their contiguity and correctness. Extensive misassemblies and NGA50 length, as determined by QUAST, are shown on the horizontal and vertical axes, respectively, in **a** and **c**. Wall clock run time (h) and peak memory (GB) usage of the genome assembly processes were recorded using the unix *time* command and shown on the horizontal and vertical axes, respectively, in **b** and **d**.

and 100.7 number of mismatches and indels per 100 kbp for HG01243, respectively, and 155.5 and 126.0, 346.1 and 379.1, and 210.2 and 103.9 number of mismatches and indels per 100 kbp for HG02055, respectively (Supplementary Tables 2, 3). The results of running BUSCO on the HG01243 and HG02055 genome assemblies can be found in Supplementary Tables 17, 18.

When substituting GoldPolish with Racon[37] for polishing the same golden path, the polishing step of the GoldRush pipeline incurred a greater computational cost, requiring over an order of magnitude more memory (602.3 vs 11.0 GB RAM) and taking 19.3% longer (9.9 vs 8.3 h) to complete, but resulted in a more base-accurate genome assembly (157.0 mismatches per 100 kbp and 106.4 indels per 100 kbp) (Fig. 3 and Supplementary Tables 19–21). The improvements in the base accuracy of the resulting NA24385 genome assembly also translated into a higher recovery of complete BUSCOs (12,752, 92.5% complete) (Supplementary Table 22). Polishing the NA24385 golden path with GoldPolish and Racon yielded QV (base quality value) statistics of 28.7 and 31.1, respectively, as assessed with Merqury (Supplementary Table 23).

To characterize the polishing performance of GoldPolish and Racon in repetitive genomic regions, we compared the resulting QV statistics of the polished assemblies, specifically looking at the statistics in non-repetitive and repetitive regions. GoldPolish polishing yielded QV statistics of 27.7 and 30.0 for non-repetitive and repetitive regions, respectively (Supplementary Table 24). Racon polishing led to QVs of 30.3 and 32.1 for non-repetitive and repetitive regions, respectively (Supplementary Table 25).

## Discussion

The GoldRush algorithm is straightforward: collect unique fragments representing the genome to generate a golden path, polish the fragments, correct them for structural misassemblies, and join the polished and corrected fragments together. As GoldRush is built upon this fundamental concept of the golden path, it represents a paradigm shift in the genome assembly of erroneous long reads, no longer requiring the time- and memory-intensive process of all-vs-all sequence alignments. Instead, the golden path, or a ~1X read fragment representation of the underlying genome, is constructed by iterating over the read set, and querying a progressive miBf database representing the golden path. We have shown that our genome assembly paradigm yields human genome assemblies that are comparable in contiguity to what can be obtained using different implementations of the OLC algorithm, yet with an order of magnitude smaller memory footprint.

The GoldRush algorithm was designed with no single long-read sequencing technology in mind, making it versatile and platform agnostic. The algorithm is robust to base errors, capable of assembling long-read datasets with estimated error rates ranging from 4 to 20%, and achieving NG50 and NGA50 lengths up to 32.6 and 22.2 Mbp, respectively, for the human data tested (Supplementary Tables 1–6).

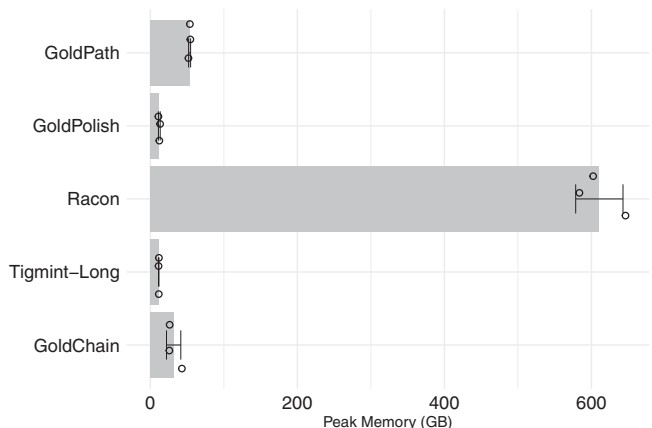

**Fig. 3 | Memory usage of the GoldRush stages when assembling human genome datasets (individuals NA24385, HG01243, and HG02055).** The average peak memory (GB) for each stage of the assembly of the three human genomes is shown, with the data points (open circles) indicating the peak memory of each assembly and the error bars indicating the standard deviation. Racon can optionally be used for long-read base polishing within GoldRush, if the onboard memory of the computer system is not limited.

GoldRush accomplishes this by mitigating the impact of base errors at various stages. For instance, GoldPath uses spaced seeds[38] to enable more sensitive detection of erroneous sequences originating from the same genomic locus. In addition, GoldPath only selects sequences with an average Phred[39,40] quality ≥15 (by default) for the silver and golden paths. The requirement ensures that the baseline assembly is composed of the higher-quality reads from the set, but may also prevent GoldPath from recruiting sequences derived from genomic loci with reads of insufficient quality. These regions, however, could be rescued downstream if GoldChain scaffolds over the region, as the gap-filling feature can recover the missing sequence. The average Phred quality threshold parameter could also be set to a lower value for more erroneous read datasets. For polishing the golden path sequences, GoldPolish accommodates the higher error rate of long reads by utilizing targeted Bloom filters[30] populated with localized $k$-mers originating exclusively from mapped reads. This approach of using a targeted Bloom filter per goldtig enables the use of smaller $k$-mer sizes than those used for the original short-read focused ntEdit+Sealer protocol, thus increasing the polishing sensitivity and mitigating errors that would otherwise arise from the use of off-target $k$-mers. Further, the GoldRush algorithm can reconstruct duplicated genomic regions in the resulting genome assembly. In our tests, we observed that GoldRush reconstructed 34.3% of duplicated human genes in multiple copies, though this figure was higher in Flye and Shasta assemblies (Supplementary Fig. 2 and Supplementary Table 14). This is also recapitulated in the Merqury spectra-cn plots;[35] the Flye and Shasta spectrum histograms more closely resemble those of the reference-grade quality NA24385 maternal and paternal genome assemblies[41] in comparison to the GoldRush and Redbean spectrum histograms (Supplementary Fig. 3). Since GoldRush uses a completely different paradigm to assemble genomes, it also has different strengths when compared to OLC-based genome assemblers. For instance, in the *O. sativa* dataset, the N50 length of the reads is 29,349 bp, and a Shasta genome assembly of the data results in an assembly NGA50 length of 104,593 bp—roughly 3.6 times longer. In contrast, the NGA50 length of the GoldRush genome assembly is tenfold greater (307.9 kbp) than the read N50 length (Fig. 2c and Supplementary Tables 4, 6). However, when considering which tool performs best on all datasets, none consistently outperforms the other in all the metrics we measured (i.e., run time, memory usage, genome contiguity, genome completeness, and genome correctness).

Yet, GoldRush is consistently more memory-efficient in comparison to other tools in all the datasets tested. This provides the opportunity to assemble long-read data from human-sized or larger genomes to those who do not have access to server-class systems, especially as retail computers with 64 GB RAM or more become more accessible. The memory efficiency of GoldRush is mainly due to the use of the miBf data structure in the GoldPath stage. However, the published version of the miBf data structure[31] was intended to serve as a static database, where the user first inserted all the items of interest, and subsequently used the database solely for query operations. For the purpose of GoldPath, we needed a memory-efficient data structure that could also be dynamic, with interleaved insert and query operations. To adjust the miBf for GoldPath, we disabled its ability to rescue information lost to hash collisions, a step that requires all the elements to be inserted, and renders the miBf static afterward. We compensated for this loss of information by using longer tile lengths, $t$. With longer tiles, there are more queries per given tile, and the increased number of queries would offset the loss of expected hits due to hash collisions.

GoldRush also assembles all the human datasets within a day and, together with Shasta, is faster than all other OLC genome assemblers tested herein. Shasta accomplishes this with heuristics based on Min-Hash markers to quickly identify potential read-to-read overlaps[20]. On the other hand, GoldRush achieves this speed with the use of a genome assembly algorithm that has linear time complexity in the number of reads (Supplementary Note 1). Breaking down the time GoldRush spends for completing each stage, we observe that GoldRush devotes more time polishing the golden path with GoldPolish (Supplementary Tables 20, 26, 27), which is already heavily optimized. GoldPolish runs a background Bloom filter building process which continually produces Bloom filters from the mapped reads for the launched pipelines, minimizing waiting time between ending and starting a new polishing run (Supplementary Fig. 4). The GoldPath and GoldChain stages were also optimized to reduce the overall run time of GoldRush. GoldPath utilizes multiple independent silver paths to effectively generate a low-coverage subsample of the original dataset, such that the subset still covers the entire target genome. With enough silver paths, any sections of the genome that are missing in a given silver path should be recovered in the others. This enables GoldPath to generate a golden path without having to process the entire dataset, speeding up the golden path generation considerably.

Further, we observed that running additional rounds of ntLink in the GoldChain stage led to substantial improvements in the contiguity of the final assembly. However, re-mapping the long reads for each round was costly. To remedy this, we implemented a mapping liftover step, allowing ntLink to run multiple rounds of scaffolding without re-mapping the long reads, thus reducing the run time in each subsequent round (Supplementary Table 28). We also implemented two additional features, overlap detection, and gap-filling, in ntLink. While these were introduced specifically for the GoldChain stage of Gold-Rush, they are also applicable to the general use of ntLink. In earlier versions of ntLink, overlapping sequences were still concatenated end-to-end, which could result in local insertion misassemblies at the contig joints. With the overlap detection and overlap resolution feature, gap estimates from the earlier stages of ntLink are used to identify putative overlaps between adjacent goldtigs (indicated by negative gap estimates), and guide trimming of the overlapping sections. Overlapping goldtigs are expected in the golden path, as reads are evaluated on a tile-by-tile basis in GoldPath.

Polishing the resulting human assemblies with GoldPolish reduces the base error rates by 53.6–85.1%. GoldPolish is built on ntEdit+Sealer[29], a $k$-mer-based base correction algorithm, and requires $2k-1$ consecutive high-quality $k$-mers to correct base mismatches and indels. This requirement is geometrically harder to meet for datasets with higher error rates. In comparison, each of the genome assemblies generated by the Flye and Shasta algorithms has higher base-pair

accuracy (Supplementary Tables 1–3, 6). In the Merqury spectra-cn plots, we observe that GoldRush has a greater number of unique $k$-mers not found in the high-quality short reads compared to competing tools, which is at least partially attributed to gap-filling with uncorrected read sequences (Supplementary Fig. 3).

With the recent release of the ONT Q20+ chemistry and its reported base accuracy of 99%, as well as the continual improvements in basecallers[42] and $k$-mer-based genome analysis solutions[43], we expect GoldRush to capitalize on the improvements in these frontiers, and reduce the time spent correcting base errors. Indeed, for the human GoldRush tests, GoldPolish achieved its highest base quality with the fastest run time for the NA24385 dataset, which is estimated to be the least erroneous in our tests (4%) (Supplementary Tables 6, 12, 15, 16, 20, 26, 27). For this dataset, GoldPolish displayed a polishing performance similar to that of Racon[37] (Supplementary Tables 12, 21, 23). Racon is computationally more expensive to run, due in part to the input reads and alignments being loaded into memory. In our tests, both GoldPolish and Racon correct repetitive regions better than non-repetitive regions (Supplementary Tables 24, 25).

Lastly, GoldRush is modular. Each step within GoldRush can be substituted for another tool that performs the equivalent function, such as substituting GoldPolish for Racon, allowing GoldRush to easily benefit from any future advances in the field. GoldRush also makes no assumptions about the quality of the input long reads, standing only to gain from future computing and sequencing improvements in the long-read sequencing domain.

We have demonstrated that our memory-efficient and modular long-read genome assembly pipeline, GoldRush, assembles long ONT reads into draft genomes with high contiguity, notably chromosome 18 telomere-to-telomere (HG02055) and several other chromosome arms (Supplementary Fig. 5). We also show that the genome assembly contiguities are comparable to what is achieved with current state-of-the-art tools, but GoldRush uses a fraction of the RAM, lowering the barrier to entry to human long-read genome assembly. With its modular design, memory efficiency, and robust performance in assembling large and complex genomes, we expect our assembly paradigm, and its first implementation, GoldRush, to both be beneficial to the scientific community and expand the reach of long sequencing reads.

## Methods

### GoldPath

To create the golden path, GoldPath first builds silver paths. A silver path is similar to the golden path, except it is an ~$rX$ (default 0.9X) representation of the genome. The silver paths are combined to generate a low-coverage subsample of the reads. This subsample is then used as input to GoldPath to generate the golden path.

GoldPath builds the silver paths by using a modified miBf[31], a resource-efficient probabilistic data structure, to associate spaced seed (a pattern with care and don't care positions) derived $k$-mers with the locus of the genome they are derived from. The miBf is composed of three data structures: a Bloom filter[30], a rank array, and an ID array. The Bloom filter is first initialized with $h$ (default 3) sets of spaced seed[38] derived $k$-mers from the input read set using ntHash[44,45]. Only reads that are at least $m$ bp (default 20,000 bp) long and have an average Phred[39,40] quality score higher than $P$ (default 15, representing 97% base accuracy or more, on average) are inserted into the Bloom filter. A rank array is then created to associate each set bit in the Bloom filter with a position in the empty ID array, sized based on the number of set bits in the Bloom filter.

The read set is then iteratively queried against the miBf data structure to determine if a read should be inserted into the miBf and, thus, the silver path. Once a read is processed and determined to contribute new base coverage to the silver path, the read is inserted into the miBf. The read is first split into tiles of length $t$ bp (default 1000), and $b$ (default 10) consecutive tiles are binned into one block.

The spaced seed-derived $k$-mers from one block are hashed and a unique ID associated with the block, synonymous with a genomic locus, is inserted into the position of the miBf ID array that corresponds to the spaced seed-derived $k$-mer (Supplementary Fig. 6). By binning tiles into blocks, GoldPath clusters together the spaced seed-derived $k$-mers from a genomic region of length $b \times t$. Each read that is inserted into the miBf is also saved to the silver path.

Like insertion, the querying process first splits the read into tiles of length $t$. The $k$-mers in these tiles are hashed using the same spaced seed patterns and queried against the miBf, then the associated ID hits within each tile are tallied in an ID-to-counts table. From these hits, a preliminary best ID hit is associated with each tile. The tiles are considered assigned (found) if the ID with the most hits exceeds a threshold $x$ (default 10), or unassigned (not found) otherwise (Supplementary Fig. 7). Next, we improve the accuracy of the tile's preliminary best hits by using information from the best hits of neighboring tiles along with the target tile's own ID-to-counts table (Supplementary Fig. 8 and Supplementary Method 1).

Once GoldPath determines the final assignment of all the tiles in the read, it evaluates the read with three possible outcomes: skip, insert, or trim and insert. If all the tiles in the read are already assigned, the read is skipped since it does not contribute any new base information to the silver path. If all the tiles in the read are unassigned, the read is inserted into the miBf data structure and the silver path in its entirety. Finally, if the read has a mixture of assigned and unassigned tiles, the read is trimmed such that only one assigned tile on either side of the longest stretch of unassigned tiles is retained (Supplementary Fig. 9), and the trimmed read is inserted into both the miBf and silver path.

When the silver path contains or exceeds the threshold number of bases (default *genome size* × $r$), the current silver path is finalized, and a new silver path is initialized. The silver path creation continues until the input reads are exhausted or the number of completed silver paths reaches $M$ (default 5). The golden path is then generated using the same algorithm and parameters used for the silver paths, except that the input is now the concatenated sequence files from the silver paths and the golden path is only complete when GoldPath has finished iterating through all of the silver path sequences. Sequences in the golden path construe the contigs of the initial genome assembly to be refined in the later steps of the pipeline, and are henceforth termed "goldtigs".

### GoldPolish

After the golden path is built, the goldtigs are polished to correct mismatches and indels. The polishing protocol, GoldPolish, closely follows the ntEdit+Sealer protocol[29], which has been shown to perform well using short sequencing reads. Unlike the currently published paradigm, which stores all the short-read $k$-mers in a single Bloom filter[30] used for correction, GoldPolish uses a targeted approach where each goldtig has a dedicated Bloom filter, each containing a hash representation of $k$-mers derived from long-read subsets.

To accomplish this, GoldPolish maps the long reads to goldtigs using, by default, minimap2[34]. GoldPolish is also capable of using mappings from other tools, such as ntLink[7]. For each goldtig, the set of mapped reads are $k$-merized using a range of different $k$-mer lengths in order to benefit from a trade-off between specificity (longer $k$-mers) and sensitivity (shorter $k$-mers). These $k$-mers are then inserted into an array of Bloom filters, one Bloom filter for each $k$-mer size per goldtig. From this, we have a set of $k$-mers targeted to individual goldtigs that we can use with ntEdit[46] and Sealer[47] for polishing (Supplementary Fig. 10).

GoldPolish then polishes the goldtigs using the full ntEdit+Sealer pipeline on each individual goldtig using their dedicated Bloom filters. GoldPolish launches multiple ntEdit+Sealer pipelines in parallel to amortize the overhead introduced with each polishing run.

## GoldChain

After polishing and correcting the goldtigs, an updated version of the long-read genome scaffolder ntLink[7] is used to assemble the goldtigs. To utilize the long-read evidence in building longer sequences, the full long-read set is mapped to the goldtigs using a lightweight minimizer-based approach. Briefly, minimizer sketches are generated for the goldtigs as well as each read for a given $k$-mer size $k$ and window size $w$[7]. The goldtig minimizer sketches are indexed, and for each minimizer in the sketch of a given long-read, this index is queried to find hits between the long read and the goldtigs. Long-read mappings that span multiple goldtigs provide scaffolding evidence. This long-read evidence is stored as a scaffold graph, where the nodes are goldtigs, and the directed edges between the nodes represent evidence that the goldtigs should be joined. This scaffold graph is traversed using abyss-scaffold[48], a heuristic-based scaffold layout algorithm, to output the final, contiguated genome assembly.

Three important features have been added to ntLink (v1.3.0+) to adapt the functionality for the de novo long-read genome assembly problem in GoldRush: overlap detection, gap-filling, and scaffolding rounds based on the liftover of sequence mappings.

For the sequence pairs with putative overlaps, minimizer sketches are generated with a lower $k$ and $w$ than the initial ntLink pairing stage to increase the sensitivity of overlap detection (parameters *small_k*, *small_w*, defaults 15 and 10, respectively). These sketches are filtered to retain minimizers that fall in the estimated overlapping region with a multiplicity of one in each sequence, ensuring that only non-repetitive minimizers in the sequences are retained. These minimizers are then used to create an undirected minimizer graph, similar to methods employed by the reference-guided scaffolder ntJoin[49]. In this graph, the minimizers are nodes, and edges between the minimizers indicate that the minimizers are adjacent in at least one of the ordered sequence minimizer sketches, with the edge weights indicating the number of sequences that have that minimizer adjacency. This minimizer graph is filtered to retain edges with a weight of 2, which removes branches and results in a graph consisting of linear path components. Each linear path is a minimizer-based mapping between the putatively overlapping sequence ends. The middle minimizer from the longest mapping is chosen to anchor the sequences to one another, and the coordinates of this minimizer guide the trimming of the detected overlapping regions on the incident sequences. Finally, after this trimming, the sequences are concatenated (Supplementary Fig. 11).

The second major feature added to ntLink uses the mapped long reads to fill gaps between the scaffolded goldtigs. The verbose option in the initial ntLink pairing stage was updated to output the complete long-read mapping information, including the mapped goldtigs and the minimizers (including position and strand on the goldtig and read). For each sequence join induced by ntLink, the verbose mapping information is parsed to identify each read that supports that join, and the associated mapped minimizers ("pass 1 minimizers"). The read with the highest average number of mapped pass 1 minimizers is chosen and subsequently used to fill the scaffold gap. As finding anchoring minimizers as close to the sequence ends as possible is preferred, each chosen read is re-mapped to the flanking sequences using a lower $k$ and $w$ for increased sensitivity ("pass 2 minimizers"). If the mapping is unambiguous, the anchoring pass 2 minimizers closest to the sequence ends are used as cut points for the flanking sequences and the read sequence filling the gap. Otherwise, the pass 1 minimizers are used to determine the gap-filling coordinates (Supplementary Fig. 12). Gap-filling is turned on in GoldRush by default and is run when the target "gap_fill" is specified to the ntLink command.

Finally, liftover-based rounds were integrated into the ntLink code base. We added a step to liftover the mapped minimizer coordinates in the verbose mapping file (described above) from the initial goldtigs to the sequences post-scaffolding. This new mapping file is then input to the ntLink pairing stage in the subsequent ntLink round, which uses the input mapping coordinates instead of re-mapping the reads to infer the scaffold graph. The remaining steps in the ntLink pipeline then proceed as previously described. To invoke these liftover-based rounds, we provided a Makefile "ntLink_rounds", which runs a specified number of rounds of ntLink (parameter *rounds*, default 5), lifting over the mapping coordinates between each iteration.

## Implementation

The GoldRush pipeline is driven by a Makefile. GoldPath and Gold-Polish are coded in C++, and GoldChain is coded in Python. All components of GoldRush utilize the btllib common code library[50]. The tool can be installed from GitHub or using the conda package manager. Instructions on how to run the GoldRush pipeline are provided on the GitHub page (https://github.com/bcgsc/goldrush). Many of the Gold-Rush parameters are supplied with default values and can be configured. Only the genome size of the target species and the long reads in a single, uncompressed, multi-FASTQ file are required as input.

## Evaluation

To evaluate the performance of GoldRush (v1.0.0), we assembled five genomes from ONT long-read data for three human cell lines (NA24385, HG01243, and HG02055), *O. sativa*, and *S. lycopersicum* (Supplementary Table 6). We optimized the parameters of GoldRush for each dataset (Supplementary Figs. 13–17 and Supplementary Table 29). In a separate trial, we also polished the golden paths with Racon[37] (v1.5.0) instead of GoldPolish. To assess the polishing performance of GoldPolish and Racon in repetitive and non-repetitive regions of the genome, we first masked the repeats in the assemblies using RepeatMasker[51] (v4.1.2) (-e ncbi -species human). Then, to generate an assembly where the non-repetitive regions are masked, we used the complement of the masked repetitive regions. These masked assemblies were then used to compare the polishing performance of GoldPolish and Racon in the repetitive and non-repetitive genomic regions. To compare the performance of GoldRush to current state-of-the-art long-read genome assemblers, we assembled all five datasets with Flye, Redbean, and Shasta. We ran both Flye (v2.9) and Redbean (v2.5) using their default parameters, and Shasta (v0.10.0) using the Nanopore-Plants-Apr2021.conf configuration file for *O. sativa* and Nanopore-May2022.conf for the other datasets.

All assemblies were analysed using QUAST[32] (v5.0.2) (--fast --large --scaffold-gap-max-size 100000 --min-identity 80 --split-scaffold), and the corresponding reference genome (Supplementary Table 30). To assess the contiguity and correctness of the assemblies, we report the NG50 and NGA50 length metrics, and the number of extensive misassemblies (as defined by QUAST). The NG50 length statistic describes that 50% of the genome size is in sequences of NG50 length or longer. The NGA50 length statistic is similar to the NG50 length, but uses alignment blocks instead of sequence lengths for the calculation. To assess the base qualities of the various assemblies, we report the number of mismatches or indels per 100 kbp from QUAST, spectra-cn plots, and QV from Merqury[35] (v1.3.0) − the latter a proxy for the log-scaled probability of error for the consensus base calls − using short reads and reference-grade genome assemblies as a comparison (Supplementary Table 31). We also ran BUSCO[33] (v5.3.2) using the primates_odb10 lineage to assess the completeness of the human assemblies in the gene space. Finally, to measure the presence of duplicated genes found in the assemblies, we used the asmgene utility in minimap2[34] (v2.24) (min coverage = 0.99 and min identity = [0.90, 0.99]) using all cDNA sequences annotated in the GRCh38 human reference from Ensembl[52] (release 87) and the T2T-CHM13[36] (v1.1) genome assembly as the "ground truth" for which genes are considered duplicated. All benchmarking tests were performed on a server-class system with 144 Intel(R) Xeon(R) Gold 6254 CPU @ 3.1 GHz with 2.9 TB RAM.

**Reporting summary**

Further information on research design is available in the Nature Portfolio Reporting Summary linked to this article.

## Data availability

The GoldRush, Flye, Redbean, and Shasta genome assemblies generated in this study have been deposited in Zenodo at https://doi.org/10.5281/zenodo.7884681[53]. The GoldRush genome assemblies generated for the parameter sweep experiments in Supplementary Figs. 13–17 are available upon request. The accession codes or location of sequencing data used for assembling the draft genomes are listed in Supplementary Table 6. The accession codes of the reference genomes and the short-read dataset used to benchmark GoldRush and comparators' genome assemblies are provided in Supplementary Tables 30–33.

## Code availability

GoldRush (v1.0.0) has been deposited in Zenodo at https://doi.org/10.5281/zenodo.7884291[54]. GoldRush is available at https://github.com/bcgsc/goldrush and released under the GPL-3 license.

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

## Acknowledgements

This study is supported by the Canadian Institutes of Health Research (CIHR) [PJT-183608, I.B.]; and the National Institutes of Health [2R01HG007182-04A1, I.B.]. The content of this article is solely the responsibility of the authors and does not necessarily represent the official views of the National Institutes of Health or other funding organizations. The funding organizations did not have a role in the design of the study, the collection, analysis and interpretation of the data, or in writing the manuscript.

## Author contributions

I.B. and R.L.W. conceived the study. J.W., L.C., and V.N. implemented the algorithms. J.W., L.C., E.Z., and P.S. analysed the data. K.M.N. provided input on the design of the algorithm. J.W., L.C., and V.N. created figures and tables with input from co-authors. J.W., L.C., R.L.W., and I.B. wrote the manuscript. R.L.W. and I.B. supervised the research. All authors commented on the manuscript.

## Competing interests

The authors declare no competing interests.
