## [Peer Review File · Nature Communications]

Linear time complexity de novo long read genome assembly with GoldRushREVIEWER COMMENTS

Reviewer #1 (Remarks to the Author):

Review of Wong et al.

The authors present a method, GoldRush, for de novo assembly. The general approach of the method is to identify reads that overlap unique regions of the genome (gold path), and to later scaffold contigs according to alignments to the unique reads. The method is fast relative to existing approaches, though with lower contiguity. Overall, the method is very well presented and unlike many reviews I do, there is no list of minor comments that need to be addressed.

There are two main comments that are short but may take some time to address accurately, both having to do with an increased stringency for measuring accuracy of assembly contiguity and base level accuracy.

The idea of avoiding all pairwise overlaps by finding unique “golden path” contigs is clever, and makes a lot of sense for genomes that are not repetitive. However, a critical concern is that repetitive DNA is explicitly not mapped using the golden path or even worse, rearranged regions may be present. The gap in NGA50 between GoldRush and Flye/Shasta is significant in humans and *S. lycopersicum*. The authors should measure the extent that duplicated DNA (duplicated genes) is present using the high quality genome <https://www.nature.com/articles/s41586-022-05325-5> .

The results of GoldPolish should be more comprehensive compared to what is possible via Racon. Again, the contigs from the HG002/NA24385 assembly can be used as ground truth. The contigs output by GoldPolish can be mapped to the Jarvis et al assembly and differences are very likely to be errors in the GoldPolish assembly. The same can be done for RACON.

Many/most researchers would be willing to invest the computational resources of RACON to get a better quality assembly. There is a concern the ntEdit+Sealer modules could restore some coding sequences but do not correct repetitive DNA.

Reviewer #2 (Remarks to the Author):

This paper presents a novel genome assembly approach, by which to assemble genomes from long reads. The approach is rooted in new data structures that (some of) the authors presented earlier, and which are based on Bloom filters at their core, with additional helper structures that aid in dealing with the identification of spaced k-mers in texts (here: reads).

The new approach is striking insofar as the main memory of the new approach are smaller by approximately one order of magnitude than previous approaches. Therefore, today, Goldrush appears to be the only approach by which one can assemble an entire human genomes on a a 64 GB RAM machine.

The approach has certain drawbacks as to the accuracy of the assemblies in terms of erroneous single nucleotide content. However, the authors argue that the approach will be helped by the increasingly accurate, and decreasingly erroneous long reads. I am ready to accept this argument. The approach will only become better with time passing, which is not necessarily the case for previous approaches. Therefore, the authors have not only presented a very resource-friendly, but also sustainable approach.

I generally support the publication of this manuscript. However, I do have a few major comments, which I think the authors need to address, before I can fully agree on publishing this manuscript. These comments refer to the embedding of the approach in the landscape of related

approaches as well as proofs for theory-natured claims that the authors raise in their manuscript, and which I feel, have been left missing.

MAJOR?

* There is no mentioning of haplotype-aware assembly, much of which is wrapped around the OLC paradigm as well (e.g. <https://doi.org/10.1186/s13059-021-02512-x>). I think it is a fair request to include such recent work into the motivating/introductory discussions or even in the comparisons, because this is something that has been keeping the community busy most recently.

Putting Goldrush into perspective with haplotype-aware assemblies makes sense for a reader who is interested in a more general comparison of what is currently possible. It also clarifies what the added benefits of Goldrush really are, and avoids confusion.

* "that runs in linear time in the number of reads" -> If this is meant to speak theoretical computer science, then it should do good if one proves this, as I feel.

I am somewhat worried about the 'GoldChain' step, because this seemingly requires to compare all long reads with all goldtigs. It would be great if you could clarify this in a more explicit runtime analysis. (what's the runtime required for querying the minimizer sketch index, for example)

I think that a formally explicit runtime analysis, for example included in the Supplement will strengthen the paper in terms of its (strong) claims.

Further, as it was shown (much) earlier that scaffolding contigs is NP-complete (I mean: https://publications.mpi-cbg.de/Huson_2002_6349.pdf for example) according to a reasonably formulated problem definition (Problem 3.1 in the mentioned paper), it would be great if you could put your work into context with these classic manuscripts.

* The large amount of mismatches in the assemblies (apart from NA24385) worries me as well. Could you please comment on the high amount, as well as the discrepancy between the error rate achieved for NA24385 and those achieved for the other genomes?

MINOR:

* There are no 'Introduction' headings

* "PacBio generally produces long reads with lower base errors (<1% for HiFi), but with shorter read lengths (typically averaging 10-25 kbp)13" -> PacBio also offer other types of reads

* "rX (default: 0.9)" -> you mean $r = 0.9$ by default?

Reviewer #3 (Remarks to the Author):

I read the manuscript titled "GoldRush A de novo long read genome assembler with linear time complexity" by Wong et al. with great interest. The authors describe the algorithm and results of a new genome assembler called GoldRush. This tool uses an innovant way to build genome assemblies by using the concept of silver and golden paths, a low coverage representation of the genome. By striving away of traditional OLC algorithms, GoldRush is able to achieve high-performance metrics at high-speed and with a low memory footprint. The tool is available on Github with sufficient guidance on how to install and run it. The manuscript is well-written but I think that some informations are missing.

- Looking at the supplementary tables, I can see that the assemblies generated by GoldRush contain up to 10 times more indels and mismatches than other tools but I could not find a place where you talk about it in the manuscript. This is a critical information and that should be included in the results and in the discussion.

- Lines 75-76: the second link has been moved and no longer works

- Line 168: "GoldRush and Shasta assembled all three human genomes in less than a day (≤ 20.8 h)". This gives the impression that Shasta and GoldRush have similar running times but looking at the supplementary tables, Shasta is between two and five times faster than GoldRush. I understand that you probably meant that both tools are to run in less than a day but I would phrase it differently.

- Lines 226-228: "The algorithm is robust to base errors, capable of assembling long read datasets with estimated error rates ranging from 4% to 20%" but later lines 232, it is stated "In addition, GoldPath only selects sequences with Phred qualities ≥ 15 ". That implies that GoldRush can only handle sequences with an error rate of up to 3.1% error rate if I am not mistaken. This would mean that if the error rate is higher than this, entire regions could be absent from the golden and silver paths. Please correct me or clarify the text.

REVIEWER COMMENTS

Reviewer #1 (Remarks to the Author):

Review of Wong et al.

The authors present a method, GoldRush, for de novo assembly. The general approach of the method is to identify reads that overlap unique regions of the genome (gold path), and to later scaffold contigs according to alignments to the unique reads. The method is fast relative to existing approaches, though with lower contiguity. Overall, the method is very well presented and unlike many reviews I do, there is no list of minor comments that need to be addressed.

[Authors' response] We thank our Reviewer for their thorough review, comments, and support of our algorithm and manuscript. We have addressed and responded to their comments below.

There are two main comments that are short but may take some time to address accurately, both having to do with an increased stringency for measuring accuracy of assembly contiguity and base level accuracy.

The idea of avoiding all pairwise overlaps by finding unique “golden path” contigs is clever, and makes a lot of sense for genomes that are not repetitive. However, a critical concern is that repetitive DNA is explicitly not mapped using the golden path or even worse, rearranged regions may be present. The gap in NGA50 between GoldRush and Flye/Shasta is significant in humans and *S. lycopersicum*. The authors should measure the extent that duplicated DNA (duplicated genes) is present using the high quality genome <https://www.nature.com/articles/s41586-022-05325-5>.

[Authors' response] We thank our Reviewer for raising this important point. We quantified the reconstructed gene duplication using the *asmgene* utility from *minimap2* (Supplementary Fig. 2 and Supplementary Table 14), and observe that GoldRush reconstructed more duplicated genes in multiples copies than Redbean, but fewer than Shasta and Flye. We now report this result in the manuscript (lines 205-210). We also ran *Merqury* to compare the spectra-cn plots generated from the genome assemblies showcased in our manuscript to that of the high-quality NA24385 genome sequence (Supplementary Fig. 3). We added sentences about the gene duplication results in the Discussion section (lines 277-284).

The results of GoldPolish should be more comprehensive compared to what is possible via Racon. Again, the contigs from the HG002/NA24385 assembly can be used as ground truth. The contigs output by GoldPolish can be mapped to the Jarvis et al assembly and differences are very likely to be errors in the GoldPolish assembly. The same can be done for RACON.

[Authors' response] We agree that using NA24385/HG002-specific data when comparing the polishing accuracy of GoldPolish and Racon is important. We also used *Merqury* to compare the base qualities of the NA24385 GoldRush assemblies using GoldPolish or Racon for the polishing step in the revised manuscript (lines 229-231 and Supplementary Table 23). We observe that QV, an estimate of base quality, is higher for Racon, which implies a more base-accurate genome assembly with respect to the high-quality NA24385 short reads used to generate the *k*-mer

database used by Merqury. This recapitulates the observations seen in the BUSCO analysis and the number of mismatches and indels statistics from QUAST.

Many/most researchers would be willing to invest the computational resources of RACON to get a better quality assembly. There is a concern the ntEdit+Sealer modules could restore some coding sequences but do not correct repetitive DNA.

[Authors' response] We thank our Reviewer for bring up this point. To investigate how well the two polishers perform on repetitive and non-repetitive genomic regions, we first masked the repetitive regions of the genome assemblies using RepeatMasker. Next, we took the complements of these masked regions to instead mask the non-repetitive regions of the genome assemblies. We then ran Merqury on the four resulting genome assemblies and observed that both Racon and GoldPolish correct both the repetitive and non-repetitive regions similarly well, and that both correct the repetitive regions better than non-repetitive regions. We report this result in our revised manuscript (lines 232-237 and Supplementary Tables 24 and 25) and discussed them in the Discussion section (lines 353-357). Also, we agree that using the best polishing algorithm will be desirable, with computational resources permitting. Anecdotally, we have heard from researchers who do not have these computational resources in certain cases, and can only resort to using GoldPolish (ntEdit+Sealer) for their GoldRush assemblies (<https://github.com/bcgsc/goldrush/issues/95#issuecomment-1350727572>).

Reviewer #2 (Remarks to the Author):

This paper presents a novel genome assembly approach, by which to assemble genomes from long reads. The approach is rooted in new data structures that (some of) the authors presented earlier, and which are based on Bloom filters at their core, with additional helper structures that aid in dealing with the identification of spaced k-mers in texts (here: reads).

The new approach is striking insofar as the main memory of the new approach are smaller by approximately one order of magnitude than previous approaches. Therefore, today, Goldrush appears to be the only approach by which one can assemble an entire human genomes on a 64 GB RAM machine.

The approach has certain drawbacks as to the accuracy of the assemblies in terms of erroneous single nucleotide content. However, the authors argue that the approach will be helped by the increasingly accurate, and decreasingly erroneous long reads. I am ready to accept this argument. The approach will only become better with time passing, which is not necessarily the case for previous approaches. Therefore, the authors have not only presented a very resource-friendly, but also sustainable approach.

I generally support the publication of this manuscript. However, I do have a few major comments, which I think the authors need to address, before I can fully agree on publishing this manuscript. These comments refer to the embedding of the approach in the landscape of related approaches as well as proofs for theory-natured claims that the authors raise in their manuscript, and which I feel, have been left missing.

[Authors' response] We thank our Reviewer for their thorough comments and support of our manuscript. We have addressed and responded to their comments below.

MAJOR?

* There is no mentioning of haplotype-aware assembly, much of which is wrapped around the OLC paradigm as well (e.g. <https://doi.org/10.1186/s13059-021-02512-x>). I think it is a fair request to include such recent work into the motivating/introductory discussions or even in the comparisons, because this is something that has been keeping the community busy most recently.

Putting Goldrush into perspective with haplotype-aware assemblies makes sense for a reader who is interested in a more general comparison of what is currently possible. It also clarifies what the added benefits of Goldrush really are, and avoids confusion.

[Authors' response] We added a paragraph to the Introduction section of the manuscript (lines 106-110) where we mention currently available haplotype-aware long read genome assemblers and stated where GoldRush fits in the context of long read genome assembly literature (lines 111-113).

* "that runs in linear time in the number of reads" -> If this is meant to speak theoretical computer science, then it should do good if one proves this, as I feel.

I am somewhat worried about the 'GoldChain' step, because this seemingly requires to compare all long reads with all goldtigs. It would be great if you could clarify this in a more explicit runtime analysis. (what's the runtime required for querying the minimizer sketch index, for example)

I think that a formally explicit runtime analysis, for example included in the Supplement will strengthen the paper in terms of its (strong) claims.

Further, as it was shown (much) earlier that scaffolding contigs is NP-complete (I mean: https://publications.mpi-cbg.de/Huson_2002_6349.pdf for example) according to a reasonably formulated problem definition (Problem 3.1 in the mentioned paper), it would be great if you could put your work into context with these classic manuscripts.

[Authors' response] We thank for the opportunity to clarify one of our central messages. We have added a runtime analysis for each step of GoldRush, and explain how each step satisfies our "linear time in the number of reads" claim (Supplementary Note 1). In addition, we included an explicit run time breakdown of GoldChain (Supplementary Tables 28 and 32). We note that the scaffold layout algorithm used in GoldChain (i.e., abyss-scaffold) uses various heuristics to increase computational efficiency, meaning that it may not compute an optimal solution.

* The large amount of mismatches in the assemblies (apart from NA24385) worries me as well. Could you please comment on the high amount, as well as the discrepancy between the error rate achieved for NA24385 and those achieved for the other genomes?

[Authors' response] We have added additional details, namely mismatch and indel rates within each human genome assemblies, to the Results section of our manuscript (lines 211-219). Further, we discuss the possible cause of the higher mismatch and indel rates seen in the GoldRush assemblies polished with GoldPolish (lines 338-346). More specifically, we clarify that GoldPolish is a k -mer based algorithm, and requires consecutive high-quality k -mers in the dataset to successfully polish sequences.

MINOR:

* There are no 'Introduction' headings

[Authors' response] We added the "Introduction" heading to our revised manuscript (line 56).

* "PacBio generally produces long reads with lower base errors (<1% for HiFi), but with shorter read lengths (typically averaging 10-25 kbp)¹³" -> PacBio also offer other types of reads

[Authors' response] We included the read lengths and error rate for PacBio Continuous Long Reads (CLR) in the long read technology comparisons, manuscript (lines 73-75).

* "rX (default: 0.9)" -> you mean r = 0.9 by default?

[Authors' response] We clarified by changing "rX (default: 0.9)" to "rX (default: 0.9X)" in the manuscript (line 376).

Reviewer #3 (Remarks to the Author):

I read the manuscript titled "GoldRush A de novo long read genome assembler with linear time complexity" by Wong et al. with great interest. The authors describe the algorithm and results of a new genome assembler called GoldRush. This tool uses an innovant way to build genome assemblies by using the concept of silver and golden paths, a low coverage representation of the genome. By striving away of traditional OLC algorithms, GoldRush is able to achieve high-performance metrics at high-speed and with a low memory footprint. The tool is available on Github with sufficient guidance on how to install and run it. The manuscript is well-written but I think that some informations are missing.

[Authors' response] We thank our Reviewer for their interest and their valuable feedback. We have addressed and responded to their comments below.

- Looking at the supplementary tables, I can see that the assemblies generated by GoldRush contain up to 10 times more indels and mismatches than other tools but I could not find a place where you talk about it in the manuscript. This is a critical information and that should be included in the results and in the discussion.

[Authors' response] We thank our Reviewer for this comment. Our 2nd Reviewer brought up a similar point; please refer to our response to their last major comment.

- Lines 75-76: the second link has been moved and no longer works

[Authors' response] We thank our Reviewer for flagging this issue. We replaced these links, added citations of two manuscripts reporting on the new ONT chemistry, and referenced the associated error rate in our revised manuscript (line 77).

- Line 168: "GoldRush and Shasta assembled all three human genomes in less than a day (≤ 20.8 h)". This gives the impression that Shasta and GoldRush have similar running times but looking at the supplementary tables, Shasta is between two and five times faster than GoldRush. I understand that you probably meant that both tools are to run in less than a day but I would phrase it differently.

[Authors' response] We have rephrased the sentence (lines 174-175) to clarify that Shasta is the fastest genome assembler, and both Shasta and GoldRush assemble each of the three human genome assemblies within a day.

- Lines 226-228: "The algorithm is robust to base errors, capable of assembling long read datasets with estimated error rates ranging from 4% to 20%" but later lines 232, it is stated "In addition, GoldPath only selects sequences with Phred qualities ≥ 15 ". That implies that GoldRush can only handle sequences with an error rate of up to 3.1% error rate if I am not mistaken. This would mean that if the error rate is higher than this, entire regions could be absent from the golden and silver paths. Please correct me or clarify the text.

[Authors' response] We thank for the opportunity to clarify our message. The Phred quality that is used to filter out reads is the average Phred quality of the read. So, even if low quality bases are present in a given read, the read sequence may still be considered for assembly as long as the average Phred quality is greater than a user-specified value ($P \geq 15$ by default). As error rate estimates are averaged across the entire read set, the base accuracy of each read spanning a given genomic locus will vary. Further, the gap-filling mechanism in GoldChain allows GoldRush to rescue genomic loci covered by reads of insufficient quality. Finally, the Phred quality is a parameter that can be controlled based on the error rates of input datasets, and we recommend not to set it too low as the GoldPath algorithm will then have trouble distinguishing highly erroneous reads from genuine reads derived from other genomic loci. We have added the above clarifications to the Discussion section of the manuscript (lines 266-271). Lastly, we expect that as the community adopts improved sequencing chemistries and basecallers, the Phred parameter could further be tuned higher to select reads with an average error rate less than 3.1%.

REVIEWERS' COMMENTS

Reviewer #1 (Remarks to the Author):

The authors have addressed my comments.

Reviewer #2 (Remarks to the Author):

The authors have addressed all of my comments to my satisfaction. I am particularly pleased with the runtime analysis provided in the Supplementary Note 1, which clarifies the linearity of the algorithm in terms of runtime complexity.

Reviewer #3 (Remarks to the Author):

Thank you for addressing my questions and concerns.